# The Preventive Effect of Lactoferrin-Containing Yogurt on Gastroenteritis in Nursery School Children—Intervention Study for 15 Weeks

**DOI:** 10.3390/ijerph17072534

**Published:** 2020-04-07

**Authors:** Teruomi Tsukahara, Anri Fujimori, Yuka Misawa, Hirotsugu Oda, Koji Yamauchi, Fumiaki Abe, Tetsuo Nomiyama

**Affiliations:** 1Department of Occupational Medicine, School of Medicine, Shinshu University, 3-1-1 Asahi, Matsumoto, Nagano 390-8621, Japan; nomiyama@shinshu-u.ac.jp; 2Department of Preventive Medicine and Public Health, School of Medicine, Shinshu University, 3-1-1 Asahi, Matsumoto, Nagano 390-8621, Japan; anrifujimori@gmail.com (A.F.); yukam@shinshu-u.ac.jp (Y.M.); 3Food Ingredients and Technology Institute, R&D Division, Morinaga Milk Industry Co., Ltd, Zama, Kanagawa 252-8583, Japan; h-oda@morinagamilk.co.jp (H.O.); ko_yamau@morinagamilk.co.jp (K.Y.); f_abe@morinagamilk.co.jp (F.A.)

**Keywords:** lactoferrin, lactoferrin-containing yogurt, nursery school children, vomiting, absent days

## Abstract

To evaluate the effects of bovine lactoferrin (LF)-containing yogurt on gastroenteritis in nursery school children during the winter season, we conducted a randomized prospective study. A total of 1296 children were randomized into a group in which LF was provided in yogurt (LF group, *n* = 661) and a non-LF consumption group (control group, *n* = 635). The LF group was given LF-containing yogurt (100 mg/day) on all 5 weekdays for approximately 15 weeks, and the control group consumed fruit jelly instead of the yogurt. The final totals of 578 children as the LF group and 584 as the control group were analyzed. The total number of children who were absent from school due to vomiting was significantly lower in the LF group compared to the control, accounting for ≥3 days in any week: 10/234 (4.3%) vs. 49/584 (8.4%), respectively; *p* = 0.04. Regarding the relationship between absences due to vomiting and the consumption of the LF-containing yogurt, the adjusted odds ratio for absence due to vomiting was 2.48 (95% CI: 1.19–5.14) in the LF children who consumed LF-containing yogurt ≤2 days/week compared to the LF children who consumed the yogurt ≥ 3 days/week. The consumption of LF-containing yogurt (100 mg/day) for ≥3 days/week might help alleviate the symptom of vomiting in nursery school children during the winter.

## 1. Introduction

Gastroenteritis is prevalent during the winter [1], and children attending nursery school have shown a higher incidence of gastroenteritis compared to when they were at home [2,3]. Symptoms of gastroenteritis, such as vomiting and diarrhea, lead to dehydration, and children with gastroenteritis are often absent on sick leave or even hospitalized for several days [4,5]. Nursery school is a semi-closed environment in which gastroenteritis outbreaks frequently occur [6,7]. The establishment of protocols for preventing gastroenteritis is an important issue for nursery schools.

Lactoferrin (LF) is an iron-binding glycoprotein that belongs to the transferrin protein family. First isolated in 1960 from human and bovine milk [8,9], bovine and human LF are composed of 689 and 692 amino acid residues, respectively. Among the human body’s secretions, colostrum has the highest LF concentration, and this concentration is higher than that of human mature milk [10,11]. An important role of LF is as a primary defense factor against mucosal infections in the viral infectious process. Orally administered LF is changed to lactoferricin (LFcin) by pepsin digestion, and LFcin also prevents the attachment of virus particles and reduces viral infection [12].

Several mechanisms underlying antivirus activities have been reported [13,14,15,16,17,18,19]. LF and its peptides have been shown to have bifidogenic activity [20]. LF and yogurt components such as oligosaccharides have synergistic effects on the growth of bifidobacterial species, and we speculated that these antivirus and bifidogenic activities of LF-containing yogurt might stimulate the intestinal immune system and prevent viral infections. Although epidemiological studies of gastroenteritis [21,22] and diarrhea in infants [23,24] have been published, the findings regarding the relationship between the intake of LF and a reduction of the incidence and the severity of gastroenteritis in children is both insufficient and inconsistent. We conducted the present study to determine whether the consumption of LF-containing yogurt would reduce the incidence and severity of gastroenteritis among nursery school children during the winter season.

## 2. Materials and Methods

This study was conducted at nursery schools in the city of Matsumoto in Japan’s Nagano Prefecture. An explanatory meeting about the study was held in 21 nursery schools (19 municipal and two private nursery schools). The study’s purpose and protocol were explained to the parents of the children at each nursery school, and informed consent for their children’s participation was obtained from 1305 parents as legal representatives of their children aged 3–6 years. The exclusion criteria were as follows: children with a milk allergy or serious present illness, the family’s plan to change residence, and a physician’s judgment for any reason. Nine children met exclusion criteria, and a final total of 1296 children participated.

We designed a questionnaire survey to obtain the children’s fundamental information: sex, age, body height, body weight, hand-washing and gargling habits, past medical history, present illness, and allergy and vaccination profiles. We randomized the children into two groups: the children who would be given bovine LF-containing yogurt (hereafter, ‘LF yogurt’; the LF group, *n* = 661) and the children who would not be given the LF yogurt (the control group, *n* = 635). The children in the LF group were given LF yogurt (100 mg/day) from Monday to Friday from 28 November, 2014 to 20 March, 2015 (approximately 15 weeks). Their parents recorded whether they consumed the yogurt or not each day and submitted the record sheets every month. The children in the control group were given fruit jelly as a substitute for the yogurt, but they were not told that they had to eat the fruit jelly.

We obtained the understanding and cooperation of The Medical Association of the City of Matsumoto, and a pediatrician provided a thorough explanation of each child’s medical examination results to the child’s parents. The only examination cost (i.e., for the gastroenteritis norovirus) was covered by research funds. The parents recorded the diagnostic results and their child’s symptoms when they consulted the pediatrician. The parents also provided their children’s medical examination reports to the children’s homeroom teachers. At the end of the study, we obtained information about the reasons for each absence from school from the respective nursery schools. The study’s protocol was approved by the Ethics Review Committee of Shinshu University School of Medicine (approval code, 3017). The trial was registered in the University Hospital Medical Information Network (UMIN) Clinical Trials Registry in Japan (registration no. UMIN000039115).

For an evaluation of the preventive effect of LF yogurt on infectious diseases, we statistically analyzed the numbers of absentees and the numbers of absent days. The absentee rates, as an index of the incidence of gastroenteritis, were analyzed with the chi-square test or Fisher’s exact test. The number of absent days, as an index of the severity of gastroenteritis, were analyzed with an unpaired t-test or the Mann–Whitney U-test. We performed a multiple logistic regression analysis to determine the odds ratios with a 95% confidence interval (95% CI) for the associations between the absence due to vomiting as a dependent variable and the independent variables of the participant’s sex, age, hand-washing habit [25] and LF yogurt consumption. A *p*-value <0.05 was considered significant. All analyses were conducted using the Statistical Package for Social Sciences (SPSS) ver. 22.0 by IBM (SPSS, Chicago, IL, USA).

## 3. Results

A total of 1296 children from 21 nursery schools participated: 661 children in the LF group and 635 children in the control group (Figure 1). The parents of 22 children (LF, *n* = 10; control, *n* = 12) withdrew the consent to participate during the follow-up period, and 1274 children participated in all 15 weeks of the study. A total of 1162 children (LF, *n* = 578; control, *n* = 584) responded properly to the essential parameters, and their data were analyzed. In the LF group, the frequencies of LF yogurt consumption in any of the 15 weeks were as follows: 5 days/week, *n* = 9 children (1.6%), 4 days/week, *n* = 44 children (7.6%), 3 days/week, *n* = 181 children (31.3%), and <3 days, *n* = 344 children (59.5%).

Table 1 summarizes the characteristics of the 1162 participants. There were no significant differences in the groups’ sex or age distribution, average height or weight, hand-washing habit, or the vaccination rate of Rotavirus.

Table 2 shows the numbers of absentees and absent days in the LF and control groups. There were no significant differences in the numbers of absentees or the number of absent days between the LF and control groups regarding each reason for absences. We defined ’all absences due to illnesses’ as the total number of absences that were due to the following: influenza, Norovirus gastroenteritis, gastroenteritis, infectious gastroenteritis, diarrhea, vomiting, common cold, fever, cough, abdominal pain, chickenpox, slapped cheek disease, *streptococcus pyogens* infection, general malaise, ear disease, nose disease, and pharyngeal pain, etc. The number of absences due to an external wound was excluded from the total amount in the analyses.

Table 3 provides the numbers of absentees and the number of absent days in the LF group (yogurt consumed ≤2 days/week), LF group (yogurt consumed ≥3 days/week), and control group. The number of absentees due to vomiting among the children who consumed the LF yogurt ≥ 3 days/week was significantly smaller than that in the control group: 10/234 (4.3%) vs. 49/584 (8.4%), respectively; *p* = 0.04. The number of days absent for all absences except external wounds among the children who consumed the LF yogurt ≥3 days/week was significantly smaller than that of the control group (3.6 ± 2.8 days vs. 4.2 ± 3.3 days, *p* = 0.02).

Table 4 is a summary of the relationships between the absences due to vomiting and the LF yogurt consumption, as revealed by the logistic regression analysis. Adjusted odds ratios (ORs) were calculated after adjustments for sex, age, hand-washing habit, and LF yogurt consumption. There was strong collinearity between the hand-washing habit and gargling (data are not shown), and we thus excluded gargling habit as an independent variable. The adjusted ORs for absences due to vomiting were 2.03 (95% CI: 1.01–4.09) in the control group and 2.48 (95% CI: 1.19–5.14) in the children taking LF yogurt ≤2 days/week compared to the children consuming LF yogurt ≥3 days/week.

## 4. Discussion

The results of our present analyses demonstrated that eating 100 mg/day of yogurt containing LF at ≥3 days/week contributed to a reduction in the number of nursery school absentees due to vomiting and reduced the number of days of absence due to illnesses other than an external wound, compared to a control group. The prevention of absences due to vomiting was significantly associated with the habitual consumption of LF yogurt after controlling for the children’s sex, age, hand-washing habit, and LF yogurt consumption. Eating LF yogurt ≥3 days/week might have alleviated the symptom of vomiting. 

Vomiting is the primary and dominant symptom of infectious enteritis [26]. The norovirus genotypes of GII.2 and GII.4 are common in Japan, and vomiting is the major symptom of the GII.2 and GII.4 norovirus [1,27]. However, our results were not consistent with these references. We might have considered that almost all the causes of vomiting were due to others except for Norovirus. LF can bind to receptors on host cells and reduce the endocytosis of microorganisms into host cells [28]. In food-borne infections caused by an enteric virus, LF interferes with the infection of cells [12] and prevents enteropathogen attachment to the intestinal epithelium and colonization [13]. LF and LFcin act against infection through viral attachment and replication [12,29]. In another study, we observed a synergistic effect of LF yogurt on the growth of bifidobacterial species [20]. These gastrointestinal effects of LF and LFcin may have alleviated the primary symptom of vomiting in the present study. Furthermore, our yogurt contained LF and also lactic bacteria. Habitual yogurt consumption and food intake, including lactic bacteria, have been reported to alleviate gastrointestinal symptoms [30,31,32,33,34,35]. Yogurt containing lactic bacteria without LF might affect intestinal microbiota composition. On the other hand, daily intake of lactic bacteria was reported to result in no change of the microbiota composition [36], no difference in the number of days of diarrhea and vomiting [37], and no effect on the prevention of diarrhea [38]. Based on the above mentioned, LF, lactic bacteria, or the synergistic effects of LF and lactic bacteria might have alleviated the symptom of vomiting.

Although the effectiveness of LF for vomiting is still being investigated, Orisaka et al. indicated that LF has an antiemetic activity, as it has a haloperidol-like effect on the part of the brain that controls vomiting [39]. Egashira et al. reported that more than half the weekly consumption of bovine LF products (i.e., 100 mg/day) ameliorated the frequency and the duration of vomiting by rotaviral gastroenteritis in children < 5 years old [21]. In that study, the weekly consumption is roughly the same as the weekly dose used in our present investigation. Oda et al. also reported that the habitual intake of an LF supplement (100 mg/day) >4 days/week reduced the gastroenteritis of their patients compared to 1 day/week. Thus, the habitual consumption of LF yogurt (e.g., 100 mg/day, 300 mg/week) might have been enough to prevent the symptom of vomiting in our study population.

We observed no significant difference in the incidence of diarrhea between the LF and control groups in our study. It was demonstrated that the consumption of 1000 mg/day of LF reduced the prevalence and severity of diarrhea in children (average age ~16 years old) [23]. In the present study, the LF amount was 100 mg/day, and the dosage used by Ochoa et al. [23] was 10-fold that used herein; such a higher amount of LF or LF yogurt might have been necessary to prevent diarrhea.

The number of absent days due to illnesses was significantly lower in the LF-group children who consumed the LF yogurt ≥3 days/week compared to the control group. Our findings suggest that the LF yogurt suppressed the onset of vomiting due to gastroenteritis. However, once the vomiting occurred, the LF yogurt did not suppress the vomiting’s severity, as shown by the days absent. There was no significant difference in the number of absent days for each illness/reason, other than vomiting, between the LF-group who consumed the LF yogurt ≥ 3 days/week and the control group. Although the number of absent days in the LF yogurt ≥ 3 days/week group was not significantly small, it was smaller than that of the control group; influenza [40,41,42], diarrhea [23,24], gastroenteritis, norovirus, common cold [24], and fever. As a result of the total amount of absent days, the number of all absent days due to illnesses might have been comprehensively reduced.

Our study has some limitations. First, we could not provide LF-free yogurt to the control group; therefore, we could not conclude whether LF, lactic bacteria, or the synergistic effects of LF and lactic bacteria alleviated the symptom of vomiting compared with LF and control group. In order to evaluate the effectiveness of LF itself, we conducted the randomized, double-blinded, placebo-control study in the next study [43]. Second, the parents of the participants may have had a higher health consciousness because they decided to let their children participate in this study. Our results might not, therefore, be generalizable to a general population. In addition, other healthy habits and behavior might have affected the two groups’ results. If random sampling is performed for all children of all nursery schools in a defined region, we could gather participants with varying levels of health consciousness and other health issues. Nevertheless, it is meaningful that there was a significant difference in vomiting between the LF and control groups recorded. Third, the records of the children’s reasons for absences and their symptoms were collected and submitted by their parents, sometimes based on a diagnosis from a physician and sometimes based on the parents’ observation (a subjective judgment); thus, some of the reasons for a child’s absence from nursey school were uncertain. Although it is ideal to obtain information directly from physicians or medical care insurance, we were often obliged to depend on the parents’ judgments.

## 5. Conclusions

The consumption of LF yogurt (100 mg/day) ≥ 3 days/week contributed to a reduction of the number of absentees due to vomiting and a reduction of the number of absent days due to illnesses other than external wounds. The habitual consumption of LF yogurt may contribute to the prevention of illness for absences in nursery schools.

## Figures and Tables

**Figure 1 ijerph-17-02534-f001:**
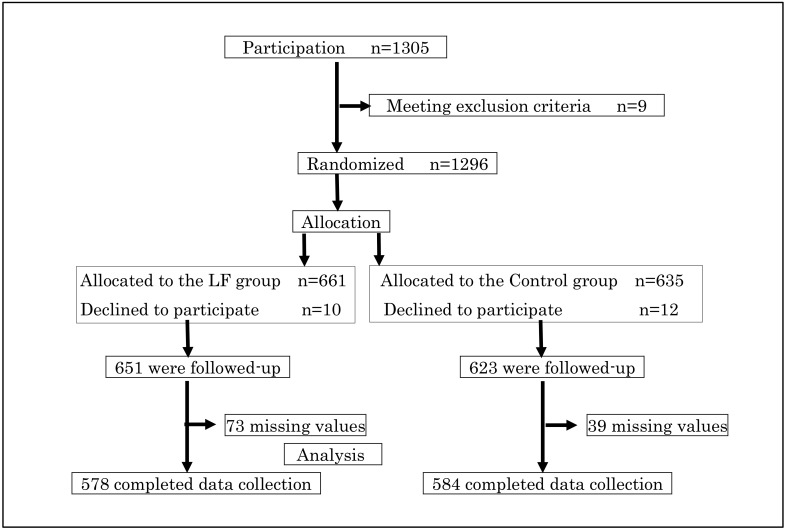
This figure is the study profile. Children wishing to participate were 1305, and 9 children were excluded, and 1296 children were randomized into the LF group or the control group. Finally, 578 LF of the group and 584 of the control group were statistically analyzed.

**Table 1 ijerph-17-02534-t001:** Characteristics of 1162 participants consisting of 578 LF group and 584 Control group.

Characteristics	All	LF Group	Control Group	*p* Value
Number of participants	1162	578	584	
Sex				
Boys n (%)	632 (54.4)	318 (55.0)	314 (53.8)	
Girls n (%)	530 (45.6)	260 (45.0)	270 (46.2)	0.67 ^a^
Age (years) Mean ± SD	4.7 ± 1.0	4.7 ± 0.9	4.7 ± 1.0	0.86 ^b^
3 years n (%)	132 (11.4)	63 (10.9)	69 (11.8)	0.90 ^a^
4 years n (%)	350 (30.1)	178 (30.8)	172 (29.5)	
5 years n (%)	408 (35.1)	205 (35.5)	203 (34.8)	
6 years n (%)	272 (23.4)	132 (22.8)	140 (24.0)	
Height (cm) Mean ± SD	107.4 ± 7.1	107.2 ± 7.3	107.6 ± 6.9	0.37 ^c^
Weight (kg) Mean ± SD	17.8 ± 3.0	17.7 ± 3.1	17.9 ± 3.0	0.23 ^c^
Hand-washing				
Habitually	1042 (89.7)	514 (88.9)	528 (90.4)	0.44 ^a^
Not habitually	120 (10.3)	64 (11.1)	56 (9.6)	
Rotavirus vaccination n (%)	13 (1.1)	4 (0.7)	9 (1.5)	0.17 ^a^

*p* value; ^a^, ^b^, and ^c^ are Chi-square test, Mann-Whitney U test and unpaired *t*-test respectively. SD; standard deviation.

**Table 2 ijerph-17-02534-t002:** The numbers of absentees and absen days in the LF group and the Control group.

Reasons for Absence	All	LF Group	Control Group	*p* Value
Number of participants	1162	578	584	
Norovirus gastroenteritis				
the total number of absentees n (%)	15 (1.3)	7 (1.2)	8 (1.4)	0.81
absent days (days) Mean ± SD	2.2 ± 1.1	2.0 ± 0.8	2.4 ± 1.4	0.81
Gastroenteritis				
the total number of absentees n (%)	43 (3.7)	27 (4.7)	16 (2.7)	0.08
absent days (days) Mean ± SD	2.1 ± 1.1	1.9 ± 1.1	2.5 ± 1.0	0.08
Infectios gastroenteritis				
the total number of absentees n (%)	29 (2.5)	14 (2.4)	15 (2.6)	0.87
absent days (days) Mean ± SD	2.0 ± 1.1	2.1 ± 0.9	1.8 ± 1.3	0.18
Diarrhea				
the total number of absentees n (%)	31 (2.7)	17 (2.9)	14 (2.4)	0.57
absent days (days) Mean ± SD	1.7 ± 1.1	1.5 ± 0.7	1.9 ± 1.4	0.67
Vomiting				
the total number of absentees n (%)	93 (8.0)	44 (7.6)	49 (8.4)	0.63
absent days (days) Mean ± SD	1.5 ± 0.8	1.6 ± 0.8	1.5 ± 0.8	0.96
All absence due to illnesses				
the total number of absentees n (%)	892 (76.8)	447 (77.3)	445 (76.2)	0.65
absent days (days) Mean ± SD	4.1 ± 3.0	4.0 ± 2.8	4.1 ± 3.2	0.77

*p*-value; Chi-square test or Fisher’s exact test were used to compare proportions between Control group and LF group in the number of absentees. Mann-Whitney U test were used to compare averages between Control group and LF group in absent days. SD; standard deviation.

**Table 3 ijerph-17-02534-t003:** The numbers of absentees and absent days in LF(<3 days/week) group, LF(3 days/week≤) group and Control group.

Reasons for Absence	3 Days or More Than 3 Days Per Week	*p* Value ^a^	2 Days or Less Than 2 Days Per Week	*p* Value ^b^	Control Group	*p* for Trend
Number of participants	234		344		584	
Norovirus gastroenteritis						
the total number of absentees n (%)	3 (1.3)	1	4 (1.2)	1	8 (1.4)	0.92
absent days (days) Mean ± SD	1.7 ± 0.6	0.5	2.3 ± 1.0	0.86	2.4 ± 1.4	0.6
Gastroenteritis						
the total number of absentees n (%)	11 (4.7)	0.16	16 (4.7)	0.12	16 (2.7)	0.09
absent days (days) Mean ± SD	1.7 ± 0.9	0.06	2.1 ± 1.2	0.23	2.5 ± 1.0	0.06
Infectios gastroenteritis						
the total number of absentees n (%)	2 (0.9)	0.17	12 (3.5)	0.42	15 (2.6)	0.5
absent days (days) Mean ± SD	2.0 ± 1.4	0.74	2.2 ± 0.9	0.16	1.8 ± 1.3	0.21
Diarrhea						
the total number of absentees n (%)	9 (3.8)	0.26	8 (2.3)	0.95	14 (2.4)	0.39
absent days (days) Mean ± SD	1.3 ± 0.5	0.45	1.8 ± 0.9	0.94	1.9 ± 1.4	0.44
Vomiting						
the total number of absentees n (%)	10 (4.3)	0.04	34 (9.9)	0.43	49 (8.4)	0.25
absent days (days) Mean ± SD	1.3 ± 0.5	0.57	1.6 ± 0.9	0.75	1.5 ± 0.8	0.88
All absence due to illnesses						
the total number of absentees n (%)	175 (74.8)	0.67	272 (79.1)	0.31	445 (76.2)	0.91
absent days (days) Mean ± SD	3.6 ± 2.8	0.02	4.3 ± 2.9	0.19	4.1 ± 3.2	0.17

*p*-value; Chi-square test or Fisher’s exact test were used to compare proportions between Control group and LF groups in the number of absentees. Mann-Whitney U test were used to compare averages between Control group and LF groups in absent days. *p*-value ^a^; between Control group and LF(3 days/week≤) group. *p*-value ^b^; between Control group and LF(<3 days/week) group. *p* for trend; Mantel-Haenszel test for trend (the number of absentees), Jonckheere-Terpstra trend test (absent days). SD; standard deviation.

**Table 4 ijerph-17-02534-t004:** Relationships of the absence due to vomiting and LF-containing yogurt intake, by the logistic regression analysis.

Variables	Crude Odds Ratio				Adjusted Odds Ratio		
	**n**	**Odds ratio**	**95 % CI**	***p* Value**	**Odds ratio**	**95 % CI**	***p* Value**
Sex							
Boys	632	1.30	0.84–2.00	0.24	1.34	0.87–2.07	0.19
Girls	530	reference			reference		
Age							
3 years	132	1.14	0.58–2.21	0.71	1.06	0.55–2.08	0.86
4 years	350	0.50	0.27–0.91	0.02	0.49	0.27–0.89	0.02
5 years	408	0.72	0.42–1.22	0.22	0.71	0.41–1.22	0.21
6 years	272	reference			reference		
The habit of hand-washing							
Habitually	1042	reference			reference		
Not habitually	120	0.80	0.38 –1.70	0.57	0.76	0.36–1.62	0.49
LF-containing yogurt intake							
Control Group	584	2.05	1.02–4.12	0.044	2.03	1.01–4.09	0.047
LF group < 3 days	344	2.46	1.19–5.08	0.02	2.48	1.19–5.14	0.02
LF group ≥ 3 days	234	reference			reference		

Adjusted odds ratio and 95% confidence interval (CI) for the absence due to vomiting based on LF group and Control group. Adjusted for sex, age, hand-washing and LF-containing yogurt intake.

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
