# Peer review of "The Preventive Effect of Lactoferrin-Containing Yogurt on Gastroenteritis in Nursery School Children—Intervention Study for 15 Weeks"

_ijerph, 2020, doi:10.3390/ijerph17072534_

Round 1

Reviewer 1 Report

Reviewers' comments:

This manuscript by dr. Tsukahara and co-workers “The preventive effect of lactoferrin-containing yogurt on gastroenteritis in nursery school children – Intervention study for 15 weeks - ” examines the effect of bovine lactoferrin-containing yogurt on gastroenteritis in children. The study was carried out in 1296 children from 21 nursery schools during the winter season.

The children were randomized into two groups: the children who would be given lactoferrin containing yogurt (661 children, lactoferrin group) and the children who would not be given the lactoferrin yogurt (635 children, control group). The children in the lactoferrin group were given lactoferrin-containing yogurt (100 mg/day, from Monday to Friday, for about 15 weeks). Parents of lactoferrin group recorded whether they consumed the yogurt or not each day and submitted the record sheets every month. The children of the control group were given fruit jelly as a substitute for the yogurt, but they were not told that they had to eat the fruit jelly.

The parents of 22 children retracted the consent to participate during the follow-up period, so 1274 children participated in all 15 weeks of the study. The data of the 1162 children (578, lactoferrin group; 584, control group) responding to the indispensable parameters were analysed. The population of children examined was optimal since there were no significant differences in the groups' sex or age distribution, average height or weight, hand-washing habit, or the rotavirus vaccination rate.

In the lactoferrin group, the frequencies of lactoferrin yogurt consumption during the 15 weeks were: 5 days/week, 1.6%; 4 days/week, 7.6%; 3 days/week, 31.3%; <3 days, 59.5%.

General Comments:

The paper is interesting and well written, but has an important flaw.

The strong limitation of this work is that the children in the control group should be given lactoferrin-free yogurt.

Numerous studies have examined the benefits of probiotics in gastroenteritis, and several clinical studies have found that probiotics can shorten the duration of diarrhoea (for example: Szajewska et al., 2001; Binns et al., 2007; Fox et al., 2015; Sur et al., 2011; Nakamura et al., 2019) so it is difficult to evaluate the activity of lactoferrin on its own also because lactoferrin has a prebiotic effect.

  • Binns CW, Lee AH, Harding H, Gracey M, Barclay DV. The CUPDAY Study: prebiotic-probiotic milk product in 1-3-year-old children attending childcare centres. Acta Paediatr. 2007; 96(11):1646–50. Epub 2007/10/17. https://doi.org/10.1111/j.1651-2227.2007.00508.x PMID: 17937689.
  • Fox MJA, K.D.K. Robertson L.K. Ball M.J. Eri R.D. Can probiotic yogurt prevent diarrhoea in children onantibiotics? A double-blind,randomised,placebo-controlled study. BMJ open. 2015; 5(e006474). https:// doi.org/10.1136/bmjopen-2014-006474 PMID: 25588782
  • Nakamura M, Hamazaki K, Matsumura K, Kasamatsu H, Tsuchida A, Inadera H; Japan Environment and Children’s Study Group. Infant dietary intake of yogurt and cheese and gastroenteritis at 1 year of age: The Japan Environment and Children's Study. PLoS One. 2019 Oct 7;14(10):e0223495. doi: 10.1371/journal.pone.0223495. eCollection 2019.PMID: 31589650
  • Szajewska H, Kotowska M, Mrukowicz JZ, Armanska M, Mikolajczyk W. Efficacy of Lactobacillus GG in prevention of nosocomial diarrhea in infants. J Pediatr. 2001; 138(3):361–5. Epub 2001/03/10. https:// doi.org/10.1067/mpd.2001.111321 PMID: 11241043.
  • Sur D, Manna B, Niyogi SK, Ramamurthy T, Palit A, Nomoto K, et al. Role of probiotic in preventing acute diarrhoea in children: a community-based, randomized, double-blind placebo-controlled field trial in an urban slum. Epidemiol Infect. 2011; 139(6):919–26. Epub 2010/07/31. https://doi.org/10.1017/S0950268810001780 PMID: 20670468.

So, in the referee's opinion, the control used is not suitable.

Specific comments:

Results showed that both the number of absentees due to vomiting and the number of days absent for all absences except external wounds among the children who consumed the LF yogurt ≥3days/week was significantly smaller than that in the control group. However there was no significant difference in the number of absent days for each illness/reason other than vomiting between the LF-group who consumed the LF yogurt ≥3 days/week and the control group.

The results obtained are not such as to support the conclusions: The habitual consumption of LF yogurt may contribute to the prevention of gastroenteritis and help reduce the outbreaks of infectious diseases in nursery schools.

lines 145-147. Authors claims “Vomiting is the primary and dominant symptom in infectious enteritis. The norovirus genotypes of GⅡ.2 and GⅡ.4 are common in Japan, and vomiting is the major symptom of the GⅡ.2 and GⅡ.4 norovirus”. However in Table 3 is shown that there is not correlation between the total number of absentees n (%) for norovirus gastroenteritis and vomiting.

Reviewer 2 Report

Brief description: Effects of lactoferrin-containing yogurt on gastroenteritis in nursery school children during the winter season were analyzed in a randomized prospective trial. Data indicated that consumption of lactoferrin-containing yogurt (100 mg/day for ≥ 3 day/weeks) reduced the number of absentees and days of absence to school for vomiting caused by gastroenteritis.

Comments to authors

Highlight: study provides robust evidence about the beneficial outcome of the consumption of lactoferrin-containing yogurt (100 mg/day, ≥ 3 day/week) in the amelioration of symptomatic gastroenteritis in children. The study has high clinical impact in the pediatric field regarding nutrition, gastroenteritis prevention and treatment. The manuscript is very well conducted, congratulations for authors.

Minor points for addressing were including in the PDF attachment

Reviewer 3 Report

This document is similar to a previous one from the same research group (DOI: 10.2147/CCID.S228153) in which the possible health benefits of the use of lactoferrin are shown. In general, the manuscript contains a substantial amount of data that are logically well presented in tables and an effective diagram. The data were obtained from a large group of people to ensure their statistical significance, and the authors honestly comment on the limitations of the study to be taken into account by readers.

The document is well written and understandable to read. The bibliography has been carefully compiled in the format requested by the journal.

In my opinion, there are not major or minor points that should be clarified.

Round 2

Reviewer 1 Report

The reviewer is satisfied with the replies and review of the article.